# Arterial Complications in Patients Undergoing Liver Transplantation After Previous TACE Treatment

**DOI:** 10.3390/jcm14041262

**Published:** 2025-02-14

**Authors:** Sebastian Weiße, Karim Mostafa, Julian Andersson, Jan-Paul Gundlach, Thomas Becker, Jost Philipp Schäfer, Felix Braun

**Affiliations:** 1Department of Internal Medicine, University Medical Center Schleswig-Holstein, Campus Kiel, 24105 Kiel, Germany; Sebastian.weiße@uksh.de (S.W.); felix.braun@uksh.de (F.B.); 2Department of Radiology and Neuroradiology, University Medical Center Schleswig-Holstein, Campus Kiel, Arnold-Heller-Street 3, 24105 Kiel, Germanyjostphilipp.schaefer@uksh.de (J.P.S.); 3Department of General and Transplantation Surgery, University Medical Center Schleswig-Holstein, Campus Kiel, 24105 Kiel, Germany; jan-paul.gundlach@uksh.de (J.-P.G.); thomas.becker@uksh.de (T.B.)

**Keywords:** TACE, arterial intervention, liver transplantation, hepatocellular carcinoma, arterial complication

## Abstract

**Introduction**: Curative treatment of HCC can be achieved by liver transplantation. In the framework of transplantation, add-on transarterial chemoembolization (TACE) can be performed as bridging therapy for local tumor control. The association between TACE and an increased incidence of hepatic arterial complications after transplantation has been investigated in multiple research items; however, the exact association remains unclear. The aim of this report was to explore the role of pre-transplantation TACE and pre-existing vascular celiac pathologies on the occurrence of postoperative hepatic arterial complications. **Methods**: This retrospective single-center study included all patients who underwent liver transplantation between 2008 and 2020. Arterial complication was defined as any postoperative occlusion, stenosis >50%, dissection or aneurysm on cross-sectional imaging. **Results**: This study encompasses 109 patients after transplantation, of which 80 underwent TACE prior to transplantation. The overall incidence of postoperative arterial complications did not differ between the groups (TACE 8/80 vs. control 6/29, *p* = 0.19). Further analysis showed no significant differences in the occurrence of specific complications (Occlusion: TACE 9/80 vs. control 3/29, *p* = 0.56; Stenosis: TACE 4/80 vs. control 5/29, *p* = 0.05; Dissection: TACE 1/80 vs. control 1/29; *p* = 0.46). Furthermore, linear regression analysis for preoperative TACE therapy, anatomic variants and pre-existing pathologies of the hepatic vasculature showed no association with postoperative arterial complications. **Conclusions**: Preoperative TACE therapy showed no influence on the incidence of post-transplant arterial complications in patients after liver transplantation. Furthermore, preoperative TACE therapy as well as anatomic variants and pre-existing arterial pathologies of the celiac axis could not be identified as risk factors for complications at the arterial anastomotic site after transplantation.

## 1. Introduction

Curative treatment of hepatocellular carcinoma (HCC) can be achieved through liver transplantation. Herein, transplantation accounted for 11.5% of all indications for this procedure in 2023 [1]. For patients on the waiting list for transplantation, add-on transarterial chemoembolization (TACE) can be performed as bridging therapy for local tumor control or tumor downstaging prior to transplantation [2]. During the TACE procedure, chemotherapeutics (e.g., doxorubicin) combined with different embolic agents are administered directly into the tumor-supplying arterial vasculature. Up until today, no unified protocol for procedural administration of TACE exists and details of the performance of the procedure vary between centers and interventionalists. In addition, there is no clear recommendation as to how many TACE procedures should be conducted prior to transplantation. However, elementary to the TACE procedure is ensuring the highest possible accumulation of chemotherapeutics and embolic agents in the tumor vasculature to maximize therapeutic effects while minimizing non-target embolization, independent of the mode of application [3]. Known complications after TACE include pancreatitis, gastritis and cholecystitis, whereas these side effects are due to non-target embolization and may be avoided by working as selectively as possible. Furthermore, choosing the most suitable access site in combination with adequately planning the embolization prior to and during the procedure, e.g., by using 3D cone-beam CT angiography imaging for navigation, may further help in avoiding non-target embolization [4].

In the past decade, it has been hypothesized that TACE itself and alterations in the hepatic arterial vasculature occurring in the framework of TACE have a negative impact on outcomes after liver transplantation, especially concerning the occurrence of complications at the site of the arterial anastomosis. Microscopic damage and inflammation of the intima and subintimal tissue seen in explanted livers have been mentioned in this framework as potential correlates for TACE-associated vascular damage [5,6,7]. Retrospective studies and a few meta-analyses have explored this topic; however, the exact influence of pre-transplantation TACE finally remains unclear, whereby some studies suggested a negative impact of TACE on arterial complications after transplantation while others could not reproduce this effect [5,6,8,9,10]. Thus, the present study aims to further evaluate this subject through a retrospective analysis of patients after liver transplantation with or without prior TACE.

## 2. Materials and Methods

### 2.1. Study Definitions

The definition of “Arterial Complication” in this study is the following: radiologically evident visual arterial stenosis of 50% or greater, pseudoaneurysm, thrombosis or occlusion of the hepatic vasculature distal of or at the level of the arterial anastomosis with corresponding liver perfusion deficits on the first CT or MRI imaging after transplantation. Anatomic variants of the hepatic vasculature were separately assessed following the Michels classification on preoperative CT or MRI imaging or interventional angiography studies [11]. Pre-existing pathologies of the hepatic vasculature described any stenosis of 50% or greater, dissection or aneurysm of the normal or anatomically variant hepatic vasculature and the celiac trunk on preoperative CT imaging.

### 2.2. Population

All patients who underwent liver transplantation in the framework of the curative treatment of HCC at our center between 2008 and 2020 were retrospectively included in this analysis, regardless of previous TACE. Baseline demographic and clinical data were gathered by chart review and are listed in Table 1.

### 2.3. TACE Procedure

All indications for TACE therapy were evaluated and finally set in an interdisciplinary board meeting involving hepatologists, surgeons and interventional radiologists after careful assessment of the available sonographies, CT or MRI imaging findings, LIRADS classifications, clinical statuses of the patients, and their potential eligibility for transplantation. In all cases, indication for neoadjuvant TACE therapy was set in the framework of bridging therapy or tumor downstaging for patients waiting for transplantation for curative treatment of HCC. In this cohort, all TACE procedures were performed by or under the supervision of a total of five interventionalists. The TACE procedure was conducted as follows. Arterial groin access was established using a short 5-French sheath. In selective cases, left-sided transbrachial or radial access was used with a long 5-French sheath. The celiac trunk was probed with a hook-shaped catheter (e.g., 5-French USL 2, RIM or cobra catheter (Boston Scientific, Boston, MA, USA)). Afterwards, a Y-valve was mounted to the hook-shaped catheter, and subsequently, a 2.7-French microcatheter (e.g., Progreat, Terumo, Tokyo, Japan) was advanced into the common hepatic artery. Next, power injector-driven angiography and late-arterial-phase contrast-enhanced cone-beam CT imaging (cbCT) were conducted to allow for a clear depiction of lesions and a 3D view of the intrahepatic arterial vasculature, allowing the identification of tumor-feeding arteries and branches (Figure 1). Additionally, selective angiography, and in selected cases, selective contrast-enhanced cbCT via the right and left hepatic artery were obtained for guiding the TACE procedure. Based on the conducted 3D angiography imaging, clear navigation into the tumor-supplying arteries was made possible and allowed for precise planning of the embolization procedure. Depending on the lesion size, configuration and associated vasculature on cbCT imaging, TACE was conducted in all cases with doxorubicin combined with degradable starch microspheres (DSM-TACE), drug-eluting beads (DEB-TACE) and conventional lipiodol TACE (c-TACE), whereby the final decision for the mode of embolization was left at the interventionalist’s discretion. In patients with singular HCC, the microcatheter was advanced as deep as possible inside the liver vasculature and TACE was performed from as close to the lesion as possible with slow continuous TACE application via hand injection and constant monitoring of antegrade flow under fluoroscopic control. In situations of multiple lesions, the microcatheter was either placed in the most reasonable selective position to minimize non-target embolization, in all cases beyond the main trunk of the right or left hepatic artery or in the distal part of the left or right hepatic artery, as unselective organ-based chemotherapy. Depending on the lesion sizes and numbers, 40–120 mg of doxorubicin was administered per TACE session. The procedure was set to be repeated every six to eight weeks after regular follow-up CT or MRI imaging to assess treatment response. TACE therapy was continued until transplantation.

### 2.4. Imaging in the Framework of Transplantation

HCC was diagnosed on multi-phase (unenhanced, late arterial, venous and delayed) contrast-enhanced CT or MRI imaging following the ACR LIRADS requirements, in selected cases with additional contrast-enhanced sonography or tumor biopsy [12]. Prior to TACE, CT imaging was conducted for planning the interventional procedure and transplantation to assess the numbers and sizes of lesions and potential variants or pathologies of the arterial and venous hepatic vasculature. In the framework of TACE, cbCT image acquisition was completed with intra-arterial contrast injection in the arterial contrast phase following a standardized protocol. In the immediate postoperative period after transplantation, point-of-care duplex sonography of the hepatic arteries was conducted to check for any pathology of the transplanted organ and the arterial, venous or biliary anastomosis. In cases of a suspected pathology, multi-phase contrast-enhanced CT was performed [13]. Further cross-sectional imaging follow-up with CT or MRI was conducted every three to six months for the following two years for assessment of the transplanted organ and potential HCC recurrence within 24 months after surgery.

### 2.5. Statistical Analysis

Data were processed using Microsoft Office and R and are presented as the mean (range) or median. Analysis of continuous and categorical variables was conducted with the Wilcoxon-Test, Mann–Whitney U Test and Fisher Exact Test as indicated. Multifactor linear regression analysis was performed for the identification of risk factors of arterial complications, whereby analysis was performed considering prior TACE, pre-existing arterial pathologies and anatomic variants classified by the Michels classification as potential risk factors [11]. The level of significance was set at an alpha = 0.05. Time-to-event data were analyzed with Kaplan–Meier statistics for the occurrence of arterial complications.

## 3. Results

### 3.1. Population

This study included a total of 109 patients who underwent liver transplantation in the setting of attempted curative HCC treatment. Demographics and baseline clinical characteristics are listed in Table 1. Overall, 80 patients underwent TACE prior to transplantation. The mean number of TACE procedures prior to transplantation averaged 4 procedures per patient, ranging from 1 to 23 TACE procedures. Parts of this cohort have been reported in previous research projects on HCC treatments [13,14,15].

### 3.2. Vascular Findings—Anatomic Variants and Pre-Existing Pathologies of the Hepatic Vasculature

Following the Michels classification, 83 patients (76.2%) were rated as Michels 1, twelve patients as Michels 2, six patients as Michels 3, one as Michels 4 and 6, respectively, and two as Michels 9 and 10, respectively [11] (Table 2). When added together, we found anatomic variants in 23.8% of our cohort.

Overall, 11/109 patients (10%) were diagnosed with a pre-existing pathology of the hepatic vasculature prior to transplantation, whereby nine of these were seen in the TACE group and two in the control group, but the difference was not significant (Table 3, *p* = 0.72).

In total, 9/80 (11.3%) pre-existing conditions were seen on pre-interventional CT imaging in the TACE group, namely five cases of celiac stenosis, three celiac aneurysms and one aneurysm of the right hepatic artery. In the control group, 2/29 (6.9%) pre-existing vascular conditions were seen, including one common hepatic artery stenosis and one celiac aneurysm (Figure 2, Figure 3 and Figure 4).

### 3.3. Analysis of Types and Occurrences of Arterial Complications

An arterial complication after transplantation was diagnosed in 14/109 patients (12.8%), whereby 8/80 (10%) were seen in the TACE group and 6/29 (20.6%) in the control group. The difference between the groups was not significant (Table 4; *p* = 0.19). Overall, eight cases of stenosis at the anastomotic site (8/109, 7.3%), three cases of arterial occlusion (3/109, 2.8%), two dissections (2/109, 1.8%) and one pseudoaneurysm of the arterial anastomosis (1/109, 0.9%) with concomitant infarction were found.

Among the eight arterial complications in the TACE group, we found three cases of arterial occlusion, four cases of stenosis at the anastomotic site with concomitant liver infarction and one dissection adjacent to the arterial anastomosis. Among the three occlusions, in one case, the common hepatic artery was occluded, while in two cases, the right hepatic artery was occluded. In all cases, concomitant liver infarction was seen, even if small.

The six arterial complications in the control group included five cases of stenosis at the anastomotic site with concomitant liver infarctions and one case of dissection. A detailed summary of the occurred complications is provided in Table 5.

Arterial complications occurred after a mean number of 78 (range, 2–193) days after transplantation and were diagnosed on contrast-enhanced CT and MRI imaging. The majority of arterial complications occurred within three months (90 days) of transplantation. Timing of the incidences of hepatic arterial complications did not differ significantly between the TACE and no-TACE group (Figure 5, *p* = 0.19).

### 3.4. Direct and Delayed Retransplantations

Overall, 8/109 patients had to undergo a retransplantation. Direct retransplantations (within 14 days after initial transplantation) were needed in five cases. All these cases had undergone preoperative TACE treatment. In this group, we report one arterial complication, namely a hemodynamically relevant stenosis of the arterial anastomosis as well as one anatomic variant of the hepatic vasculature (Michels 3). We report no pre-existing pathologies in this group. Finally, there was no statistically significant difference to the group of patients who did not undergo preoperative TACE concerning a direct retransplantation (5/80 vs. 0/29; *p* = 0.32). The remaining three retransplantations were performed delayed in a timeframe >14 days after initial transplantation. In this group, two patients had undergone preoperative TACE while one had not. We report one pre-existing stenosis of the celiac trunk and one pre-existing anatomical variant of the hepatic vasculature (Michels 2) in this group. Again, no significant difference between the TACE and no-TACE group was found concerning delayed retransplantations 14 days after initial transplantation (2/80 vs. 1/29; *p* = 1).

### 3.5. Analysis of Risk Factors for Arterial Complications

Binary logistic linear regression analysis was performed as indicated for the evaluation of the influences of TACE, pre-existing arterial pathologies and anatomic variants of the hepatic vasculature prior to transplantation regarding the occurrence of arterial complications after liver transplantation, whereby no statistically significant influence of any of these factors could be shown (TACE *p* = 0.24; pre-existing arterial pathology *p* = 0.13; anatomic variants of hepatic vasculature *p* = 0.62).

## 4. Discussion

TACE represents a cornerstone of minimally invasive HCC treatment, providing the possibility for local tumor control, downstaging prior to surgery and bridging therapy prior to curative liver transplantation [16,17,18]. However, concerns remain regarding arterial complications encountered in patients after liver transplantation who receive preoperative TACE treatment. Herein, in the past two decades, a multitude of retrospective studies and meta-analyses have specifically addressed this topic; however, the exact association between TACE and arterial complications after liver transplantation remains largely unclear due to conflicting evidence [5,6,9,19,20,21,22,23,24,25]. Our main findings are the following: (1) We found no statistically significant difference in the occurrence of arterial complications after liver transplantation in the TACE group vs. the control group. (2) Preoperative TACE treatment could not be identified as a risk factor for the occurrence of an arterial complication after liver transplantation. (3) Pre-existing pathologies and anatomic variants of the hepatic vasculature were not associated with postoperative arterial complications.

### 4.1. TACE and Arterial Complications After Transplantation

In our study, the overall occurrence of arterial complications after liver transplantation was 12.8% (14/109). There was no significant difference in occurrence between the TACE group and the control group (8/80 vs. 6/29; *p* = 0.19). Furthermore, we found no statistically significant differences regarding the occurrences of arterial occlusions, stenoses or dissections among the TACE group and the control group (Arterial Occlusion 3/80 vs. 0/29, *p* = 0.56; Stenosis 4/80 vs. 5/29, *p* = 0.05; Dissections 1/80 vs. 1/29, *p* = 0.46). Linear regression analysis showed no statistically significant influence of preoperative TACE, pre-existing pathologies or anatomic variants of the hepatic vasculature on the occurrence of arterial complications. In 2014, Panaro et al. reported an overall amount of 14.9% of arterial complications in 67 patients undergoing transplantation for HCC, which is comparable to our findings; however, their reported association between preoperative TACE and postoperative arterial complications could not be reproduced in our cohort. Furthermore, they reported a higher percentage of 21.9% of arterial complications in the subgroup of patients after TACE, compared to our results of a 10% arterial complication rate in the similar group [6]. A meta-analysis by Sneiders et al. from 2018 including ten studies reported a range of 2.3–21.9% (3.7–21.9% after TACE; 2.3–13.1% without previous TACE) of arterial complications in patients undergoing liver transplantation. While the results of our study fit into these ranges, we could not reproduce a significantly higher number of arterial complications in patients who underwent TACE therapy prior to transplantation [5]. In 2021, Sneiders et al. published a retrospective study of 876 patients from six European centers, reporting 6.6% of patients with hepatic arterial complications overall and 6.4% in the subgroup of patients after TACE, both considerably lower than our findings [26]. The main difference to Sneiders et al. is the higher number of anastomotic stenoses in our cohort overall (1.9% vs. 8.3%) and in the preoperative TACE group (1.9% vs. 5.0%). This discrepancy could be attributed to the aggressive cross-sectional follow-up imaging schedule with more accurate vascular diagnostics by contrast-enhanced CT in our study, while only one center of the multicenter cohort assessed patients with cross-sectional imaging after six months. As for arterial occlusion after preoperative TACE, there is a good agreement between Sneiders et al. from 2021 and our study (3.5% vs. 3.8%). Recently, Selim et al. in 2023 reported an occurrence of 6.2% of arterial complications in 162 patients after transplantation, including 6.4% in 110 patients who did not undergo TACE in comparison to 5.7% in patients who underwent TACE. While this represents a considerably lower number of complications compared to our study, it is necessary to highlight that postoperative hepatic arterial stenosis was not considered a complication in their study, which might again be attributed to our cross-sectional follow-up imaging protocol allowing for more precise vascular diagnostics. Apart from the mentioned discrepancies to Selim et al. from 2023, our conclusion of a lack of association of preoperative TACE treatment and postoperative hepatic arterial complication is in harmony with their results and those of Sneiders et al. from 2021 [25,26].

### 4.2. Direct and Delayed Retransplantations

In our study, a total of 8/109 patients had to undergo a retransplantation. Out of those, five retransplantations occurred within 14 days of the procedure and three in a delayed timeframe. We found no statistically significant difference between the TACE and control group in both the early and delayed retransplantation timeframes (TACE group *p* = 0.32; control group *p* = 1). Approximately 5–22% of patients will require a retransplantation after liver transplantation [27]. Considering preoperative TACE as a risk factor, there are currently no studies fully addressing this matter. Sneiders et al. addressed surgical and interventional reinterventions in their meta-analysis in 2021, reporting pooled results from five studies with 7.2% of reinterventions in TACE patients and 4.1% in non-TACE patients, which is slightly higher than our findings due to our study not including interventional radiological reinterventions [5,6,7,9,28,29].

### 4.3. Considerations on the Application of TACE Prior to Liver Transplantation

In theory, damage in the celiac trunk, the common hepatic artery, and both the right and left hepatic artery in the framework of TACE can occur due to either mechanical stress following probing of the intrahepatic vasculature, drug toxicity or a combination of both. It is known that TACE can lead to micro- and macroscopic damage of the arterial wall with fibrosis, edema and inflammation of the intimal and subintimal vascular layer as well as the occurrence of segmental arterial occlusions [6,7,26,28]. In our cohort, TACE therapy was conducted as selectively as possible from segment and subsegment branches of the hepatic arteries, allowing for the best possible drug and embolic agent accumulation in the tumor vasculature with minimized non-target embolization, sparing both the main and lobar trunks of the hepatic vasculature used for graft anastomosis [30,31]. Furthermore, given the superselective approach we used with slow hand injection of the embolic mixture under constant fluoroscopic control to ensure consistent antegrade flow, we could also minimize backflow alongside the microcatheter, which is likely the most common cause of non-target embolization in TACE. Different techniques have been conceived to further ensure backflow along the catheter does not occur, e.g., performance of TACE under balloon occlusion with specific microcatheters [32]. In the framework of repeated TACE procedures, macroscopic arterial alterations of the tumor-feeding vasculature, e.g., stenosis, vessel wall irregularities and segment branch occlusions may be present (Figure 1). However, on imaging, these alterations are limited to the immediate perimeter of the lesion deep inside the hepatic vasculature and will be seen only in branches affected by TACE therapy, where they do not affect the extrahepatic vasculature used for graft anastomosis. As for future research, histopathological analysis as conducted by Panaro et al. 2014 may be performed of the intrahepatic arterial segments directly receiving TACE in comparison to the extrahepatic TACE-naïve vasculature, which would require a close collaboration between radiologists, pathologists and surgeons [6].

### 4.4. Clinical Implications

In the face of the multitude of retrospective studies and meta-analyses covering the presented topic, the results of our study further support the claim that preoperative TACE therapy is not associated with postoperative occurrence of arterial complications after liver transplantation and that neoadjuvant TACE therapy should not be withheld from eligible patients on the waiting list for transplantation or in need of downstaging. TACE has to be seen as a powerful therapeutic tool in the setting of hepatocellular carcinoma. Several studies have shown the safety and efficacy of TACE therapy and have confirmed similar outcomes for patients after transplantation independent of preoperative TACE [25,33]. Furthermore, TACE may allow patients exceeding transplantation criteria to be eligible for transplantation by tumor downstaging, which presents a relevant benefit for this collective [34,35]. In a meta-analysis by Parikh et al. in 2015, it was shown that downstaging strategies including TACE can be successful in up to 50% patients. More recently, a study by Morais et al. showed a successful downstaging of 44.7% in 45 patients, whereby successful tumor size reduction was possible in 69.1% of the patients after only one TACE session [2]. Tumor response after a singular locoregional therapy, including TACE, is a known phenomenon previously reported by Mehtta et al., and the performance of a singular procedure will not affect the extrahepatic vasculature used for graft anastomosis during transplantation, given the safe performance of it as described above [36].

As per the consensus, locoregional treatments of HCC should only be considered if the waitlist period is expected to surpass 6 months; however, due to the unpredictability of organ availability nowadays, almost all centers have adopted these locoregional treatments as the standard [37]. Following the earlier mentioned positive effects of TACE, e.g., downstaging and successful bridging, the application of TACE and neoadjuvant TACE will allow for an elongation of waitlisting periods, given careful performance of the procedure and continuous tumor monitoring with CT or MRI imaging. Currently, liver transplantation represents the only curative treatment option for patients with unresectable HCC. However, despite the importance of transplantation in this collective, patients will often encounter waitlisting with unclear timeframes until transplantation following a shortage of organs. Especially in patients with end-stage liver disease, HCC patients exhibit the highest waitlisting for transplantation [38,39]. In these patients, to avoid hepatic tumor progression while waitlisted with subsequent waitlist dropout, it will remain important to offer TACE therapy as an option for bridging until transplantation. Considering the overall aging population, it can be expected that organ demands overall and for HCC patients will rise, which may further emphasize the performance and refinement of TACE and neoadjuvant TACE to allow for longer waitlist periods.

### 4.5. Limitations

This is a retrospective study with several limitations that need to be considered when interpreting the results. Given the retrospective single-center design, the number of patients was low, especially in the TACE-naïve group, and our results therefore are closely associated to local diagnostic and therapeutic practices and expertise, limiting the overall generalizability of our findings. The definitions of this study included postoperative hepatic arterial stenosis diagnosed on contrast-enhanced CT imaging as arterial complication; however, the number of these findings was still low given the sample size. While these findings were clearly depicted by CT imaging, their significance alone may be insufficient without the synopsis of clinical and laboratory parameters.

## 5. Conclusions

The incidences of post-transplant arterial complications in patients treated for hepatocellular carcinoma do not differ based on the status of preoperative TACE treatment. Furthermore, TACE could not be identified as a risk factor for complications at the arterial anastomotic site after transplantation.

## Figures and Tables

**Figure 1 jcm-14-01262-f001:**
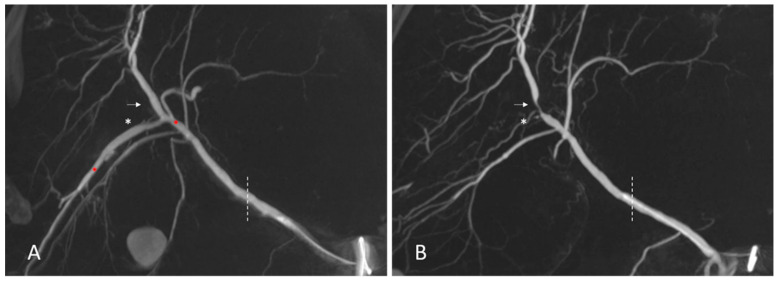
Contrast-enhanced cone-beam CT imaging depicting intrahepatic arterial alterations after eight DSM-TACE procedures. During the first TACE procedure, a variant supply of the right hepatic artery from the superior mesenteric artery is depicted. Super-selective DSM-TACE was subsequently conducted with drug application from distal branches of the right hepatic artery (**A**, red stars). After eight DSM-TACE procedures, arterial alterations like stenosis (white arrows, **B**) and segment branch occlusion (white stars, **B**) can be seen. However, all these changes are distant from the extra- to intrahepatic junction, which marks the most distal site for arterial graft anastomosis (dotted lines).

**Figure 2 jcm-14-01262-f002:**
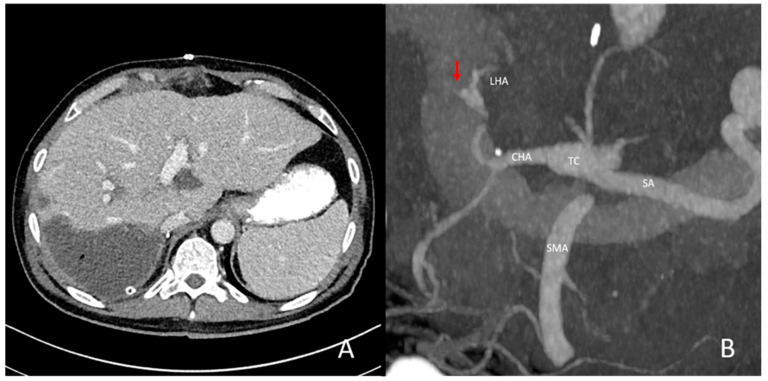
Intrahepatic occlusion of the right hepatic artery with concomitant liver infarction. In this patient, image (**A**) depicts large infarction of the right liver lobe involving segments V, VI and VII after liver transplantation. In image (**B**), maximum-intensity projection reconstruction of the celiac axis shows no contrasting of the proximal right hepatic artery (**B**, red arrow), suggesting intrahepatic arterial thrombosis.

**Figure 3 jcm-14-01262-f003:**
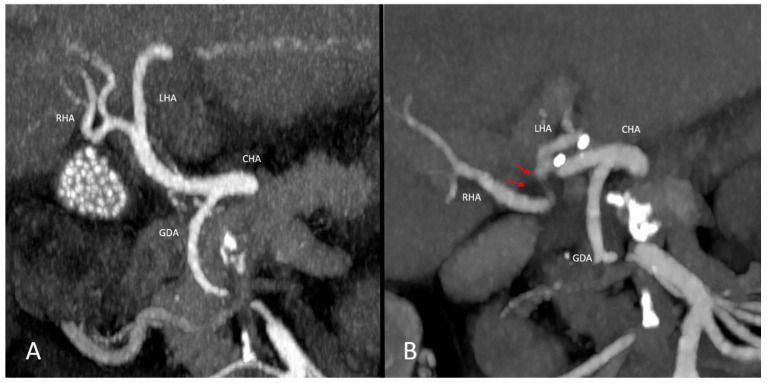
Stenosis of the propriate hepatic artery after liver transplantation. In this patient, preoperative contrast-enhanced CT imaging depicts normal hepatic vasculature (image **A**). Follow-up CT imaging after liver transplantation shows stenosis >50% of the propriate hepatic artery (red arrows, image **B**).

**Figure 4 jcm-14-01262-f004:**
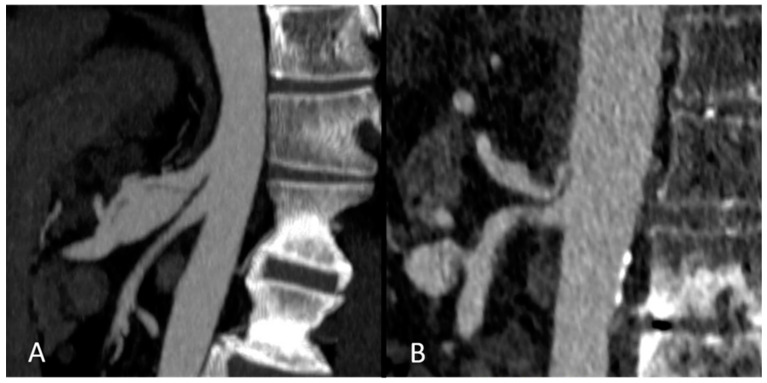
Examples of pre-existing pathologies of the celiac trunk on preoperative CT imaging. (Images **A**,**B**) show contrast-enhanced imaging prior to liver transplantation in two patients. In (image **A**), a dissecting aneurysm of the celiac trunk is depicted while (image **B)** shows a severe stenosis of the celiac trunk due to compression of the arcuate ligament.

**Figure 5 jcm-14-01262-f005:**
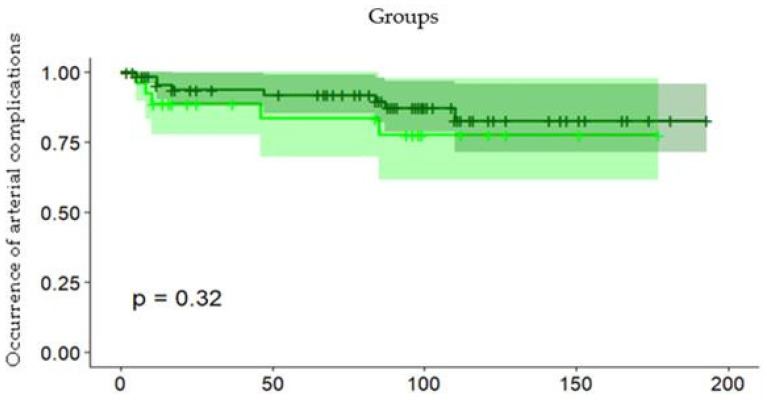
Kaplan–Meier statistics for the occurrence of arterial complications after transplantation.

**Table 1 jcm-14-01262-t001:** Demographic and HCC characteristics.

	TACE Received (80)	TACE Not Received (29)	*p*-Value
Sex (f/m) (n)	16/64 (80%)	9/20 (69%)	0.302
Mean age at LTx (yrs)	62.06 (32–75)	59.94 (45–70)	0.055
AFP-positive HCC (n)	43	9	0.050
Milan In (n)	51	15	0.275
UCSF In (n)	9	2	0.724
Mean waiting time (d)	229	165	0.747
**Predictors**			
Size of largest tumor (cm)	3.6 (1.1–14)	2.5 (0.4–11.9)	0.002
Number of lesions (n)	1.88	1.62	0.205
Vascular invasion radiologic (V+) (n)	11	3	0.755
Max. AFP value (ng/dL)	258.35	273.93	0.023
**Comorbidities**			
Diabetes (n)	33	17	0.130
pAD (n)	7	1	0.678
Hypertony (n)	43	15	1
Smoker (n)	8	4	0.729
cHD (n)	12	2	0.345
I.v. dr(n)	4	1	1
**Etiology of Cirrhosis**			
C2-toxic	28	18	0.052
Hepatitis A	3	2	0.621
Hepatitis B	17	3	0.176
Hepatitis C	31	4	0.019
Hepatitis D	1	1	0.489
Hepatitis E	0	1	0.284
NASH	8	6	0.215
Alpha-1 antitrypsin deficiency	0	1	0.284
Autoimmune-related	0	1	0.284
MELD at transplantation (median, range)	16 (7–34)	22 (6–36)	0.034
**Postop. Complications**			
CD-I	27	6	0.241
CD-II	5	4	0.242
CD-IIIa	5	4	0.242
CD-IIIb	7	3	0.724
CD-IVa	6	2	1
CD-IVb	8	4	0.729
CD-V	8	2	1
**Retransplantations**			
<14 d to Tx	5	0	0.322
>14 d to Tx	2	1	1
**Arterial Complications (n)**	8	6	0.193
Dissection (n)	1	1	0.462
Stenosis (n)	4	5	0.051
Occlusions (n)	3	0	0.561

**Table 2 jcm-14-01262-t002:** Michels classifications for both groups.

	Overall (n = 109)	TACE (n = 80)	Control (n = 29)
**Michels 1**	83 (76.1%)	61 (76.3%)	22 (75.9%)
**Michels 2**	12 (11.0%)	11 (13.8%)	1 (3.4%)
**Michels 3**	6 (5.5%)	3 (3.8%)	3 (10.3%)
**Michels 4**	1 (0.9%)	0	1 (3.4%)
**Michels 6**	1 (0.9%)	0	1 (3.4%)
**Michels 9**	2 (1.8%)	1 (1.3%)	1 (3.4%)
**Michels 10**	2 (1.8%)	2 (2.6%)	0

**Table 3 jcm-14-01262-t003:** Anatomic variants and pre-existing pathologies of the hepatic vasculature.

	Overall (n = 109)	TACE (n = 80)	Control (n = 29)	*p*-Value
**Celiac Aneurysm**	3 (2.8%)	3 (3.8%)	0	0.500
**Celiac Stenosis**	6 (5.5%)	5 (6.3%)	1 (3.5%)	1
**Hepatic Artery Aneurysm**	1 (0.9%)	1 (1.3%)	0	1
**Hepatic Artery Stenosis**	1 (0.9%)	0	1 (3.5%)	0.260
**Overall**	11 (10.1%)	9 (11.3%)	2 (7.0%)	0.720

**Table 4 jcm-14-01262-t004:** Overview of arterial complications.

	Overall (n = 109)	TACE (n = 80)	Control (n = 29)	*p*-Value
**Arterial Complications**	14 (11.7%)	8 (10.0%)	6 (20.7%)	0.193
**Arterial Occlusion**	3 (2.7%)	3 (3.8%)	0	0.561
**Stenosis**	9 (8.3%)	4 (5.0%)	5 (17.2%)	0.050
**Dissection**	2 (1.8%)	1 (1.3%)	1 (3.5%)	0.460

**Table 5 jcm-14-01262-t005:** Detailed overview of arterial complications.

Arterial Complication	Days	Management	Outcome
**TACE Group, n = 8**			
Stenosis	87	Surgical revision	Death after revision due to transplant failure
Celiac dissection	84	Conservative (ASS)	Good
Arterial thrombosis	47	Stenting of upstream celiac stenosis	Good, compensated arterial occlusion
Arterial thrombosis	12	Surgical revision	Good, intraoperative no thrombosis detected
Arterial thrombosis	5	Planned retransplant	Death prior to retransplant
Stenosis	110	Conservative	Good
Stenosis	12	Conservative	Good
Stenosis	17	Retransplantation due to severe infarction	Good
**Control Group, n = 6**			
Stenosis	137	Conservative	Chronic transplant dysfunction
Dissection	85	Conservative	Good
Stenosis	46	Retransplantation due to biliary complication	Good
Stenosis	10	Conservative	Good
Stenosis	8	Conservative	Good
Stenosis	5	Retransplantation	Multi-organ failure, death

## Data Availability

The dataset analyzed in this research is available from the authors upon reasonable request.

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
