# Peer review of "Arterial Complications in Patients Undergoing Liver Transplantation After Previous TACE Treatment"

_jcm, 2025, doi:10.3390/jcm14041262_

Round 1
Reviewer 1 Report
Comments and Suggestions for Authors
1) The authors aimed to explore the role of pre-transplantation TACE and preexisting vascular coeliac pathologies on the occurrence of postoperative hepatic arterial complications. This was a retrospective single-centre study including all patients who underwent liver transplantation between 2008 and 2020.
2) The study is novel and interesting and it is also well written. I think it is acceptable with minor revisions that may help to improve the quality of the manuscript.
3) The introduction is too short in the current format and some more information as background should be added. For example cite the recent paper by Minici R et al. Efficacy and Safety of Distal Radial Access for Transcatheter Arterial Chemoembolization (TACE) of the Liver. Journal of personalized medicine, 13(4), 640. https://doi.org/10.3390/jpm13040640
4) Methods are robust and well presented.
5) Results are relevant and well described with appropriate tables and figures.
6) Discussion section are very well written with good discussion points on TACE and arterial complications after transplantation, Direct and delayed retransplantations, Considerations on application of TACE prior to liver transplantation, and Clinical Implications. Limitations are properly acknowledged.
7) Conclusions are justified by the results.
8) Language is fine.
Author Response
Dear Reviewer 1,
Please find the answers to your comments in the attached document.
Kind regards

Reviewer 2 Report
Comments and Suggestions for Authors
The following is my critical appraisal of some positive points and strengths of the study:
This retrospective, single-center study including 109 patients who underwent liver transplantation for hepatocellular carcinoma (HCC) aimed to investigate the association between pre-transplant transarterial chemoembolization (TACE) and the occurrence of hepatic arterial complications after liver transplantation. The study conclusions were that the overall incidence of arterial complications did not differ significantly between the two groups who received TACE group: 8/80 (10%) and control group: 6/29 (20.6%) p = 0.19 (not statistically significant).
Specific complications reported were Occlusion: TACE 3/80 vs. Control 0/29 (p = 0.56); Stenosis: TACE 4/80 vs. Control 5/29 (p = 0.05); Dissection: TACE 1/80 vs. Control 1/29 (p = 0.46).
Conclusion: Preoperative TACE does not increase arterial complications risk following liver transplantation. Anatomic variants and preexisting vascular conditions are not significant risk factors for post-transplant arterial complications.
The following are six points of minor concerns
1. I would suggest that the Authors emphasize some relevant limitations such as the fact that this is a single-center, retrospective design with a small sample size, particularly the control group with only 29 patients, making statistical power limited. Some comparisons are underpowered. E.g. arterial occlusion
2. There is no mention of a strict standardized TACE protocol leading to possible heterogeneity in treatment. The study does not specify whether all patients underwent the same TACE technique (drug-eluting beads vs. conventional TACE) or whether multiple sessions were required.
3. Were all TACE patients comparable with the control group in terms of MELD or Child Pugh criteria?
4. I would suggest the authors add to in the discussion the need for further research with larger, multi-center, prospective studies to confirm their findings. Since TACE can cause arterial inflammation, fibrosis, and endothelial damage, which may contribute to complications consider discussing the possible role of a histopathological analysis of the hepatic artery in the explanted liver arteries.
5. Could it be possible for authors to include a table summarizing all similar studies (sample size, arterial complication rate, major findings)?
6. Clearly define the clinical significance of each complication in terms of management; e.g., did arterial stenosis require intervention, retransplantation?
Author Response
Dear Reviewer 2,
Please find the answers to your comments in the attached document.
Kind regards

Round 2
Reviewer 2 Report
Comments and Suggestions for Authors
All criticisms and questions raised by the reviews were satisfactorily addressed by the authors.